# Order–disorder transition of a rigid cage cation embedded in a cubic perovskite

Zhifang Shi [1], Zheng Fang[1], Jingshu Wu[1], Yi Chen[1] & Qixi Mi [1✉]

The structure and properties of organic–inorganic hybrid perovskites are impacted by the order–disorder transition, whose driving forces from the organic cation and the inorganic framework cannot easily be disentangled. Herein, we report the design, synthesis and properties of a cage-in-framework perovskite $AthMn(N_3)_3$, where $Ath^+$ is an organic cation 4-azatricyclo$[2.2.1.0^{2,6}]$heptanium. $Ath^+$ features a rigid and spheroidal profile, such that its molecular reorientation does not alter the cubic lattice symmetry of the $Mn(N_3)_3^-$ host framework. This order–disorder transition is well characterized by NMR, crystallography, and calorimetry, and associated with the realignment of $Ath^+$ dipole from antiferroelectric to paraelectric. As a result, an abrupt rise in the dielectric constant was observed during the transition. Our work introduces a family of perovskite structures and provides direct insights to the order–disorder transition of hybrid materials.

[1] School of Physical Science and Technology, ShanghaiTech University, Shanghai, China. ✉email: miqx@shanghaitech.edu.cn

Organic–inorganic hybrid perovskites cover a range of compositions and have found extensive applications as photovoltaic[1], luminescent[2,3], and ferroelectric[4–7] materials. The organic cation, normally methylammonium (MA$^+$) or formamidinium (FA$^+$), occupies the perovskite A-site, freely spins at near room temperature, behaves like a spherical cation, and interacts with the host framework through van der Waals forces and hydrogen bonding[8,9]. This host–guest interaction has a strong influence on the macroscopic structure and properties of the perovskite material. For example, the phase transition of dimethylammonium zinc triformate [DMAZn(HCOO)$_3$] from paraelectric to antiferroelectric at ~156 K is accompanied by DMA$^+$ losing its rotational freedom[10,11].

Such an order–disorder phase transition is typical for hybrid perovskites known so far[8,10,12,13]. In the low-temperature (LT) phase, the stationary organic cation induces the host framework to adopt a lower lattice symmetry. When the guest cation becomes unfrozen, the high-temperature (HT) phase experiences switching in the electric[12,14–17] and magnetic[18–21] properties. However, the contributions from the guest cation and the host framework to these property changes are generally difficult to separate from each other. A solution to this issue is to find a system whose order–disorder transition involves minimal deformation in both the guest and host, and this condition can be satisfied by a rigid and spheroidal cation.

A first candidate for the spheroidal cation would be tetramethylammonium (TMA$^+$, Table 1), which is much larger in size than MA$^+$ and requires an expanded host framework comprising polyatomic bridges. For example, the triazide perovskite structures TMAMn(N$_3$)$_3$[12,18], TMACd(N$_3$)$_3$[22], and TMACa(N$_3$)$_3$[23] are known. In their LT phases, the host framework becomes distorted to achieve close packing with the stationary TMA$^+$ bearing a tetrahedral profile. The LT phases of several bimetallic perovskites TMA$_2$M$^I$M$^{III}$(N$_3$)$_6$ (M$^I$ = Na, K; M$^{III}$ = Cr, Fe) have been found[24] to adopt a cubic structure. Nonetheless, the nonpolarity of TMA$^+$ is ineffective in inducing dielectric or ferroelectric switching during phase transitions. Another familiar spheroidal cation quinuclidinium (Q$^+$, Table 1) has not been reported to be contained in a framework. Several binary salts of Q$^+$ have been studied[25,26] to reveal the existence of an intermediate-temperature (IT) phase, where Q$^+$ rotates around its three-fold axis like an ellipsoid. Taking a lesson from the downsides of TMA$^+$ and Q$^+$, a polycyclic ammonium with 5–6 carbons, i.e., a small molecular cage, would better resemble a rigid sphere.

In this work, we identified 4-azatricyclo[2.2.1.0$^{2,6}$]heptanium (Ath$^+$, Table 1), likely the smallest stable tricyclic cation, and report its synthesis with complete NMR and crystallographic data. This cation enabled us to create a unique cage-in-framework perovskite AthMn(N$_3$)$_3$ and investigate how its structure and properties are affected by the host–guest interaction. Our results indicate that whether or not Ath$^+$ is immobilized in AthMn(N$_3$)$_3$ does not affect the cubic symmetry of the host framework, whereas an antiferroelectric–paraelectric transition of the dipole alignment of Ath$^+$ results in a significant switching effect in the dielectric spectrum.

## Results and discussion

**Cage cations**. Unlike other spheroidal organic ammonium ions in Table 1, which are commercially available, 4-azatricyclo[2.2.1.0$^{2,6}$]heptanium (Ath$^+$) has so far only been incompletely mentioned once before[27]. In Supplementary Fig. 1, we designed a multistep synthetic route to Ath$^+$. To build the tricyclic cage structure, a protected maleimide underwent Corey–Chaykovsky cyclopropanation to yield a mixture of *exo-* and *endo-* isomers, followed by reduction and deprotection, but only the *endo*-isomer had the correct configuration for closing the third cycle, see details in Supplementary Information.

Table 1 lists the ionic radii of Ath$^+$ and several other spheroidal ammonium ions, obtained using both theoretical and experimental methods. In the first method, the molecular geometry of the ammonium ions was optimized by Density Functional Theory (DFT) calculations. For each organic ammonium ion, a minimal sphere was constructed to contain all the hydrogen nuclei, and the ionic radius was considered to equal the radius of this enclosing sphere plus the Bohr radius (0.53 Å). Alternatively, single-crystal X-ray diffraction (SC-XRD) showed that the iodides of these organic ammonium ions assume the distorted NaCl crystal lattice. (Supplementary Table 1) The average anion–cation distance subtracted by the radius of I$^-$ (2.20 Å) gave the cationic radius. Likewise, the calculated and crystallographic radii of methylammonium (MA$^+$) were found to be 2.02 and 2.01 Å, respectively. In Table 1, however, Abh$^+$, Ath$^+$, and Q$^+$ exhibit experimental ionic radii smaller than predicted by calculations. This trend could be explained by a much less H to C ratio, and hence less steric hindrance, of these cage cations. For example, Ath$^+$ consists of more C and fewer H atoms than the open-chain TMA$^+$ (tetramethylammonium), but these two cations appear almost the same in size. Therefore, Ath$^+$ has a more compact structure than TMA$^+$ and could fit better in a negatively charged framework.

**Table 1 Structure and size of several spheroidal organic ammonium ions.**

| Cation[a] | TMA$^+$ | Abh$^+$ | Ath$^+$ | Q$^+$ |
|---|---|---|---|---|
| Formula | C$_4$H$_{12}$N$^+$ | C$_5$H$_{10}$N$^+$ | C$_6$H$_{10}$N$^+$ | C$_7$H$_{14}$N$^+$ |
| Calc. structure[b] | | | | |
| Calc. radius (Å) | 2.68 | 2.75 | 2.83 | 2.97 |
| Exp. radius (Å) | 2.70 | – | 2.69 | 2.77 |

[a]TMA$^+$ tetramethylammonium, Abh$^+$ 2-azabicyclo[2.1.1]hexanium, Ath$^+$ 4-azatricyclo[2.2.1.0$^{2,6}$]heptanium, Q$^+$ Quinuclidinium.
[b]Brown, gray, and pink balls stand for C, N, and H atoms, respectively, and the outer sphere visualizes the ionic radius.

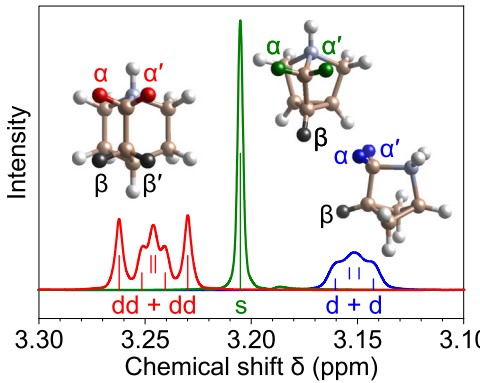

**Fig. 1 Comparison between the NMR splitting patterns of the α methylene protons of Abh⁺Cl⁻ (blue), Ath⁺I⁻ (green), and Q⁺I⁻ (red) in DMSO-$d_6$.** The α protons of Ath⁺ exhibits a singlet peak, apparently not coupled to its β proton. Vertical lines indicate peak positions obtained from curve fitting. (Supplementary Figs. 2–4).

Figure 1 plots the solution-phase NMR spectra of three polycyclic ammonium ions Abh⁺, Ath⁺, and Q⁺, focusing at $\delta = 3.1–3.3$, the range of chemical shifts corresponding to the -CH₂N⁺- group. For Abh⁺, the splitting pattern of these α methylene protons can be fitted by two overlapping doublets, indicating that the α and α′ protons are coupled differently to the β proton (coupling constants $J = 2.3$ and 9.0 Hz). The case of Q⁺ having two β protons is more complicated, where the two α protons are split into two sets of doublet of doublets (dd + dd, Fig. 1), with $J$ ranging from 5.3 to 10.8 Hz. For both Abh⁺ and Q⁺, the magnetic inequivalence between the α and α′ protons suggests lack of mirror symmetry along the chain -CH₂CH₂N⁺- and a distorted molecular geometry.

By contrast, the α protons of Ath⁺ show a sharp singlet peak in the presence of a β proton. If this peak is actually a converging doublet, lineshape fitting estimates that $J < 0.6$ Hz. (Supplementary Fig. 4) The Karplus equation predicts that $J$ vanishes only when two vicinal C–H bonds are fixed at a perpendicular dihedral angle ∠HCCH. Detailed results of $J$ in various conformations of the tetrahydrofuran ring[28] can be formulated by

$$J \approx (9\,\text{Hz}) \cdot \cos^2\left(\frac{9}{8}\angle\text{HCCH}\right) \qquad (1)$$

According to the calculated structure of Ath⁺, ∠HCCH = 65° and the above equation gives $J = 0.8$ Hz, qualitatively agreeing with the experimental value. In short, the remarkably small proton coupling constant in Ath⁺ reveals a symmetric and rigid cage structure.

**Frameworks containing a cage cation.** Guided by the trend of ionic radii in Table 1, we identified the Mn(N₃)₃⁻ framework, which forms a distorted perovskite structure with TMA⁺, to also accommodate the cage cations Abh⁺, Ath⁺, and Q⁺. Synthesis from their chloride salts was straightforward by mixing with stoichiometric MnCl₂ and LiN₃ in methanol under N₂. The precipitated crystals were confirmed by SC-XRD to be $A$Mn(N₃)₃, ($A$ = Abh⁺, Ath⁺, and Q⁺, see Supplementary Tables 2–4) with an expected cage-in-framework structure.

We studied the phase-transition behavior of $A$Mn(N₃)₃ before in-depth characterizations of their crystal structures. In Fig. 2, Differential Scanning Calorimetry (DSC) shows that AbhMn(N₃)₃ and AthMn(N₃)₃ undergo first-order phase transitions at 299–300 and 339–341 K, respectively. The transition temperature of QMn(N₃)₃ is even higher at 351 K, possibly in two continuous steps, and the reverse transition occurs in separate steps at 347 and 320 K, which suggests the existence of a metastable phase between the two temperatures. The high-temperature (HT) phase

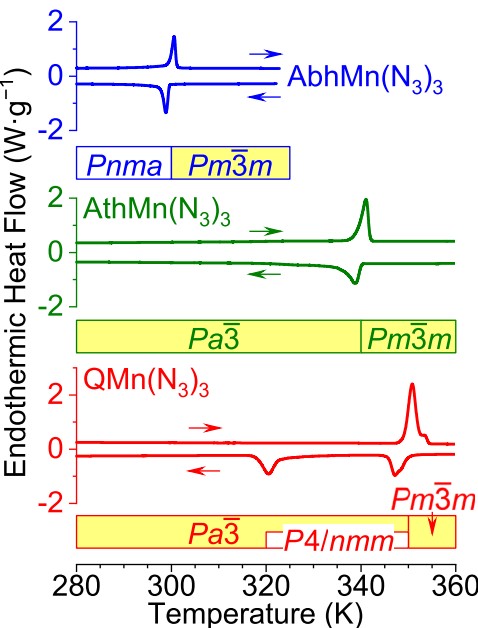

**Fig. 2 Cyclic DSC curves and phase transition diagrams for AbhMn(N₃)₃ (blue), AthMn(N₃)₃ (green), and QMn(N₃)₃ (red) between 280 and 360 K.** No phase transitions were found between 100 and 280 K for these compounds.

of all three $A$Mn(N₃)₃ adopts the cubic perovskite structure containing a randomly oriented cage cation $A^+$. When cooled down below the first transition temperature, AbhMn(N₃)₃ turns into the orthorhombic $Pnma$ space group as a result of Abh⁺ being motionless, whereas QMn(N₃)₃ loses its three-fold crystal symmetry and enters an intermediate-temperature (IT) phase. On the other hand, AthMn(N₃)₃ maintains a cubic perovskite structure throughout the temperature range 100–373 K.

From the DSC results, we estimated the entropy change of AthMn(N₃)₃ to be 15 J·mol⁻¹·K⁻¹ during its phase transition at 339–341 K. According to Boltzmann's entropy formula

$$\Delta S = R \ln \frac{N_2}{N_1} \qquad (2)$$

where $N$ is the number of accessible microstates and $R$ is the gas constant, a ratio of $N_2/N_1 = 6$ is obtained for the phase transition. This value is consistent with a simple mechanistic picture that Ath⁺ gains a disorder/order ratio of 3 and the Mn(N₃)₃⁻ matrix gains 2. By contrast, previously reported azide perovskites exhibited significantly higher $\Delta S$ values, such as ~20 J·mol⁻¹·K⁻¹ for TMACd(N₃)₃[22] and TMAMn(N₃)₃[12], or 23–26 J·mol⁻¹·K⁻¹ for TMAMᴵ₀.₅Mᴵᴵᴵ₀.₅(N₃)₃[24]. Compared with these systems, the relatively small $\Delta S$ of AthMn(N₃)₃ suggests disorder in its HT phase to be well-defined, which highlights a close match of the Mn(N₃)₃⁻ framework with the spheroidal profile of Ath⁺, more so than with other cations TMA⁺, Abh⁺, or Q⁺.

We further monitored the phase transition of AthMn(N₃)₃ at 339–341 K by solid-state NMR using the Cross Polarization and Magic-Angle Spinning (CP/MAS) technique. The ¹³C NMR spectra in Fig. 3 include two sets of ¹³C peaks at the chemical shift ranges $\delta = 55–80$ and 8–15 ppm, corresponding to the α and β carbons of Ath⁺, respectively. For the low-temperature (LT) phase of AthMn(N₃)₃ at 298 and 318 K, these two peaks displayed relatively broad Lorentzian linewidths of 1.0–2.6 ppm. The phase transition begins at 338 K, when the two broad peaks fade away and two narrow peaks ascribed to the HT phase show up. At 346 K, only the two narrow peaks of Lorentzian widths 0.3–0.4 ppm remain. Also, $\delta$ of the two ¹³C peaks move closer to each other, similar to the solution-phase

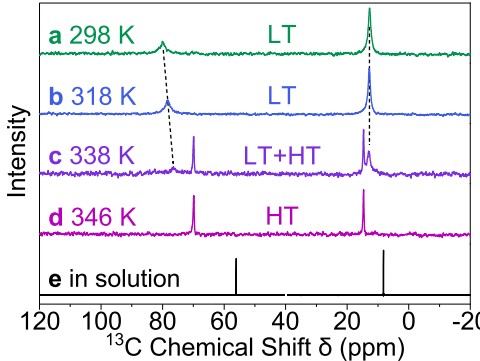

**Fig. 3 Solid-state and solution NMR spectra of AthMn(N₃)₃. a–d** Solid-state $^{13}C$ NMR spectra of AthMn(N₃)₃ showing a transition from the low-temperature (LT) to high-temperature (HT) phases. **e** $^{13}C$ NMR spectrum of Ath⁺ in DMSO-$d_6$ at 298 K with the solvent peak at 40 ppm omitted.

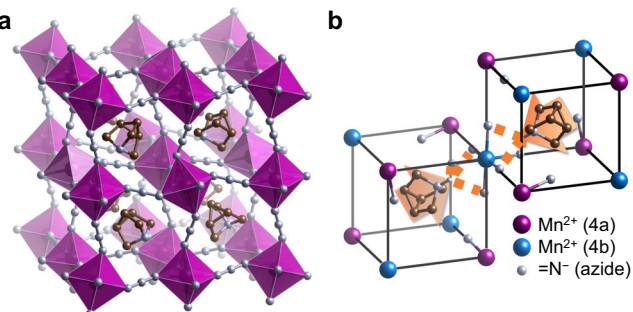

**Fig. 4 Crystal structure of the low-temperature phase of AthMn(N₃)₃ at 300 K. a** The Ath⁺ cage cation is embedded in a cubic perovskite framework made of MnN₆ octahedra (magenta) and azide linkers. **b** In each octant (black frame) of the unit cell, Ath⁺ is oriented along the body diagonal by hydrogen bonding (dotted line) with azido N, leading to an antiferroelectric alignment of the dipole moments (arrows).

spectrum of Ath⁺. These effects of phase transition on linewidth and δ signify a more isotropic environment around Ath⁺, attributable to its free spinning, in the HT phase of AthMn(N₃)₃.

**Dipole alignment in AthMn(N₃)₃.** Figure 4a depicts the single-crystal structure of the LT phase of AthMn(N₃)₃ at 300 K, which is a cage-in-framework extension of the simple perovskite structure. The MnN₆ octahedra linked by azide groups are all tilted, because the azide linker is linear but the $sp^2$-hybridized terminal N⁻ always coordinates with Mn²⁺ at an angle, forming a *trans* bridge between each adjacent pair of MnN₆ octahedra. The ISODISTORT program[29] reveals that such unconventional tilting can be represented by a mixture of the $X_5^-$ and $M_5^+$ modes having similar amplitudes[30]. Nonetheless, the octahedral centers are not shifted relative to each other[31], due to the spheroidal shape and suitable size of Ath⁺. As a result of octahedral tilting without shifts, the LT phase of AthMn (N₃)₃ adopts a cubic $Pa\bar{3}$ space group and its unit cell is composed of eight formula units. (Supplementary Table 2)

Figure 4b illustrates the geometric relationship between the ionic components of AthMn(N₃)₃. In each octant of the unit cell, there are one Ath⁺ at the A site, one Mn²⁺ at two inequivalent B sites (Wyckoff positions 4a and 4b), as well as three N₃⁻ or six terminal N⁻. These terminal N⁻ are not uniformly distributed inside an octant of the unit cell; three of them ligate the same Mn²⁺₄b, whereas each of the rest three is bonded to a Mn²⁺₄a. (Fig. 4b) On the other hand, all six N⁻ around a Mn²⁺₄b are located inside two

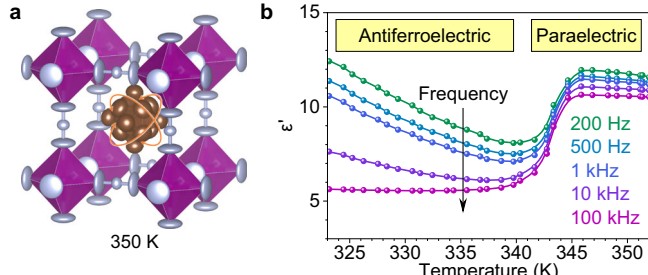

**Fig. 5 High-temperature structure and dielectric behavior of AthMn(N₃)₃. a** Thermal ellipsoid model of AthMn(N₃)₃ in the high-temperature phase at 350 K, where the Ath⁺ cage cation tumbles freely. **b** Temperature and frequency dependence of the dielectric constant $\varepsilon'$ of AthMn(N₃)₃, showing a transition from antiferroelectric to paraelectric.

opposite octants of the unit cell, because of linear ∠NMnN bond angles. As a result of simultaneously forming hydrogen bonds (~2.4 Å in length) with three N⁻ ligands, the Ath⁺ cage cation shifts slightly toward a Mn²⁺₄b, and the electric dipoles of Ath⁺ are paired up antiferroelectrically along the body diagonal of the unit cell.

At 350 K, the HT phase of AthMn(N₃)₃ belongs to the $Pm\bar{3}m$ space group with increased symmetry and a reduced unit cell. In Fig. 5a, the orientation of Ath⁺ becomes disordered, the azido N⁻ are uniformly distributed at the 6f position of the unit cell, and the preferred orientation for hydrogen bonding no longer exists. Still, it is presumable that the linear N₃⁻ maintains a *trans* configuration between adjacent Mn²⁺ at disordered orientations, represented by a nearly isotropic thermal ellipsoid in the middle and large, oblate ones for the terminal N⁻. Figure 5b plots the dielectric constant $\varepsilon'$ of AthMn(N₃)₃ as a function of temperature and frequency. At 100 kHz, $\varepsilon'$ displays a step-like increase from 5.6 to 10.6 after the phase transition from LT to HT. The LT phase responds more prominently to lower probing frequencies, typical of a slow dipolar relaxation process[32]. In the HT phase of AthMn(N₃)₃, $\varepsilon'$ is insensitive to frequency between 200 Hz and 100 kHz, which suggests a paraelectric behavior due to free tumbling of Ath⁺. This is distinct from TMAMn(N₃)₃ and its homologs[12,18], where the open-chain ammonium $(CH_3)_nNH_{4-n}^+$ at the A-site barely contributes to the dielectric behavior of the whole perovskite material. In short, the order–disorder transition of Ath⁺ serves as the main driving force for AthMn(N₃)₃ to convert between the antiferroelectric and paraelectric phases.

In summary, we designed and synthesized a polycyclic ammonium Ath⁺, and employed it to build a family of cage-in-framework structures. Ath⁺ is special for its rigid molecular backbone and spheroidal shape that best fits the Mn(N₃)₃⁻ framework, so that AthMn(N₃)₃ maintains a cubic perovskite structure throughout 100–373 K. In the LT phase of AthMn(N₃)₃, pairs of Ath⁺ are hydrogen bonded to a central MnN₆ octahedra, yielding an antiferroelectric configuration. Ath⁺ starts to spin freely at ~340 K, witnessed by solid-state NMR and SC-XRD. This transition also leads to a paraelectric alignment of the Ath⁺ dipole and an abrupt rise in the dielectric constant. Overall, thanks to the distinctive cage-in-framework structure of AthMn (N₃)₃, movement of the spheroidal Ath⁺ directly contributes to the antiferroelectric–paraelectric transition and brings about little excess entropy to the perovskite matrix. AthMn(N₃)₃ offers a quintessential model for studying the order–disorder transition and more disorder-related phenomena.

## Methods

**Spectral data of 4-azatricyclo[2.2.1.0²,⁶]heptanium iodide (AthI).** $^1H$ NMR (500 MHz, DMSO-d₆) δ: 10.27 (s, 1H), 3.21 (s, 6H), 1.90 (s, 3H). $^{13}C$ NMR (126

MHz, DMSO-$d_6$) $\delta$: 56.08, 8.21. HRMS (ESI, $m/z$): calcd for $[C_6H_{10}N]^+$ 96.0808, found 96.0811. Its NMR spectra (Supplementary Fig. 2) and single-crystal structure (Supplementary Fig. 5) are provided in the Supplementary Information.

**Synthesis and crystal growth of 4-azatricyclo[2.2.1.0$^{2,6}$]heptanium manganese(II) triazide [AthMn(N$_3$)$_3$].** In three separate vials, 0.147 g (3.00 mmol) LiN$_3$[33], 0.126 g (1.00 mmol) anhydrous MnCl$_2$, and 0.132 g (1.00 mmol) AthCl was each dissolved in 2.0 mL methanol. The three solutions were mixed, turning from colorless to light green and then to light yellow, and were passed through a 0.22 μm membrane filter. Standing for ~15 min at room temperature initiated the precipitation of AthMn(N$_3$)$_3$ particles. After the supernatant was removed using a syringe and replaced by 2.0 mL ethylene glycol, the AthMn(N$_3$)$_3$ precipitate was completely dissolved at 95 °C, and then allowed to cool down to 25 °C at 1.0 °C·h$^{-1}$ in a programmed thermostat. Light yellow, transparent cuboids of sizes up to 2 mm appeared at the bottom of the vial. (Supplementary Fig. 6)

Similarly, 2-azabicyclo[2.2.1]hexanium manganese(II) triazide [AbhMn(N$_3$)$_3$] was synthesized from 2-azabicyclo[2.2.1]hexane hydrochloride (Pharmablock PB06149-01), and quinuclidinium manganese(II) triazide [QMn(N$_3$)$_3$] from quinuclidine (Energy Chemicals W610532) and hydrogen chloride (4.0 M in 1,4-dioxane).

**Single-crystal X-ray diffraction (SC-XRD).** SC-XRD was performed on a Bruker D8 VENTURE diffractometer with Mo Kα radiation ($\lambda = 0.71073$ Å). The diffraction data were handled by the Olex2 software[34], and the crystal structures were solved and refined by the ShelXT[35] and ShelXL[36] programs, respectively. Thermal ellipsoid models were rendered by VESTA[37] or Ortep3[38]. Variable-temperature measurements were enabled by blowing a stream of N$_2$ gas at 150 to 360 K to the mounted crystal.

**Nuclear magnetic resonance (NMR).** Liquid $^1$H and $^{13}$C NMR spectra were measured on a 500 MHz spectrometer (Bruker ADVANCE). Slight distortion in the $^1$H spectral signals was corrected using reference deconvolution, before they were decomposed into individual Voigt peaks with identical lineshape parameters.

Solid-state $^{13}$C NMR was acquired on a 400 MHz spectrometer (Bruker ADVANCE III HD), with the cross polarization (CP) and magic-angle spinning (MAS at 15 kHz) techniques applied to enhance signal intensity and reduce peak broadening.

**Differential scanning calorimetry (DSC).** Warning! Metal azides are explosive to various degrees. In a poorly planned experiment, an AthMn(N$_3$)$_3$ single crystal of ~5 mg exploded at 568 K and shattered the sample holder.

DSC was performed on a PerkinElmer DSC 8000 calorimeter equipped with a liquid nitrogen cooling system. A sample of ~5 mg was weighed, sealed in an aluminum pan, and monitored under a temperature program of 298 → 380 → 100 → 360 K at a ramp rate of 10 K·min$^{-1}$ alongside an empty pan. Baseline drift was canceled by averaging the blank heating and cooling segments.

**Temperature-dependent dielectric behavior.** Capacitors made of AthMn(N$_3$)$_3$ pellets was placed in a programmed oven with the two copper electrodes connected to a precision LCR meter (Agilent E4980A). The measurements were carried out in the parallel capacitance–dissipation factor mode with an applied voltage of 100 mV and 1 kHz. The sample's relative dielectric constant $\varepsilon'$ was calculated from measured capacitance $C$ according to the following equation:

$$\varepsilon' = \frac{4Cd}{\pi\varepsilon_0 D^2} \qquad (3)$$

where $d$ and $D$ stand for the thickness and diameter of the sample disk, respectively, and $\varepsilon_0$ is the vacuum permittivity.

**Theoretical calculations.** Molecular geometries of the organic ammonium cations were optimized by Density Functional Theory (DFT) calculations, using the B3LYP functional and 6-31+g(d,p) basis set implemented in Gaussian 09[39]. The resulting Cartesian coordinates of the hydrogen nuclei were converted to a minimal enclosing sphere (Table 1) by the MinBoundSphere code[40], and ionic radius of the organic ammonium equals the radius of this sphere plus the Bohr radius (0.53 Å).

## Data availability

The data that support the findings of this study are available from the corresponding author upon reasonable request. The X-ray crystallographic coordinates for structures reported in this study have been deposited at the Cambridge Crystallographic Data Centre (CCDC), under deposition numbers 2073935–2073943. These data can be obtained free of charge from The Cambridge Crystallographic Data Centre via www.ccdc.cam.ac.uk/data_request/cif.

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

## Acknowledgements

This work is financially supported by a State Key Research Project (No. 2016YFA0204000) from the Ministry of Science and Technology of China, a research grant (No. 22075182) from the National Science Foundation of China, and funding from ShanghaiTech University. The authors thank the Analytical Instrumentation Center of ShanghaiTech University, Dr. Na Yu for assistance with single-crystal crystallography, Dr. Min Peng for solid-state NMR, and Mr. Dejun Dai for differential scanning calorimetry.

## Author contributions

Q.M. conceived and directed the project. Z.S., J.W., and Y.C. carried out organic synthesis. Z.S., F.Z., and J.W. prepared single crystals. Z.S. and F.Z. performed crystallography and various property measurements. Q.M. and Z.S. analyzed the experimental data and wrote the paper.

## Competing interests

The authors declare no competing interests.
