## [Peer Review File · Nature Communications]

REVIEWER COMMENTS

Reviewer #1 (Remarks to the Author):

Hybrid organic-inorganic perovskites are well-known compounds for their attractive properties that can be easily tuned by proper choice of organic and inorganic components. The authors report synthesis of a new azide perovskite comprising a new rigid and spheroidal organic cation. They show that the compound exhibits a structural phase transition from the Pm-3m to Pa-3 phase. This transition is associated with change of the dielectric permittivity from about 6 to about 11. I have no doubts that the data are reliable and that analysis of almost all experimental data was performed in a correct way. I have, however, impression that the authors' conclusions are overclaimed. Therefore, I am not convinced that the paper is suitable for Nature Communications. It seems more suitable for a more specialized journal, like Chemistry of Materials or Journal of Materials Chemistry C.

Specific comments:

1. Introduction: the authors wrote "This host-guest interaction has a strong influence on the macroscopic structure and properties of the perovskite material. For example, the transition of FAPbI₃ from the black perovskite phase to the yellow hexagonal phase is accompanied by FA⁺ losing two degrees of its rotational freedom." The yellow phase is not a PEROVSKITE phase. Therefore, better example would be MAPbI₃, MAPbBr₃ or recently discovered methylhydrazinium lead bromide perovskites, which undergo phase transitions from the disordered to ordered phases.

2. DSC data: the authors claim that "The cubic phase yields only a few diffraction points, insufficient for resolving the disordered A-site cation." Please add information on phase transition entropy and the calculated value of N using the formula $\Delta S = R \ln N$. This will provide information on number of configurational states of the disordered cation in the HT phase.

3. The authors wrote "the orientations of Ath⁺ and N₃⁻ become disordered" and they conclude that "AthMn(N₃)₃ offers a prime model for studying the pure order-disorder transition and more disorder-related phenomena." They also wrote in the Abstract "Our work introduces a new family of perovskite structures and provides direct insights to the order-disorder transition of hybrid materials." and "This pure order-disorder transition..."

Firstly, the phase transition leads also to tilting of the MnN₆ octahedra and I think that also to shift of the Ath⁺ cation. These are displacive contributions and therefore the phase transition is not 100% of order-disorder type. I recommend to add some information on the shift of the organic cation.

Secondly, the same symmetry change, from Pm-3m to Pa-3 associated with ordering of organic cation and N₃⁻ ligands were reported for double perovskite family (NMe₄)₂[MIIIMI(N₃)₆] (MIII=Cr, Fe; MI=Na, K) (see Z.-Y. Du et al, Cryst. Growth Design. 2014, 14, 3906-3909). Therefore, novelty of this paper is limited and the authors cannot claim that their AthMn(N₃)₃ compound is unique and that it offers "a prime model for studying the pure order-disorder transition..."

4. The authors should compare their results with those reported for the mentioned above tetramethylammonium double perovskite azides (Z.-Y. Du et al, Cryst. Growth Design. 2014, 14, 3906-3909). The disorder of the azide anions is seen through very large thermal ellipsoid of the terminal N atoms. Can the structure be better refined assuming disordered model for the azide anions, like in the case of double perovskite azides?

5. The authors wrote in the Abstract "The structure and properties of organic-inorganic hybrid perovskites are impacted by the order-disorder transition, whose driving forces from the organic cation and the inorganic framework cannot easily be disentangled." Other statements in the Abstract and the text suggest that in case of AthMn(N₃)₃ the driving force from the inorganic framework can be neglected. This sentence is a little bit misleading since although the framework structure remains cubic, this compounds still presents two driving forces that may be difficult to disentangled: one is ordering of the Ath⁺ cation and the second force is ordering of the N₃⁻ linkers. I have no doubt that the main driving force of the phase transition is ordering of the organic cation but for sure ordering of the azide anion also contributes to the phase transition mechanism.

Reviewer #2 (Remarks to the Author):

The manuscript by Shi and co-workers presents three new members of the family of molecular perovskites, in which spheroidal polycyclic ammonium cations ("Abh+", "Ath+" and "Q+") sit at the A site within manganese azide cages. Like many other members of this family, the materials undergo an order-disorder, dielectric-antiferroelectric phase transition. The authors make the particular claim to significance that the Ath+ material has a cubic space group both above and below the phase transition -- unlike the Abh+ material, which has a low-temperature orthorhombic phase, and the Q+ material, which has an intriguing metastable tetragonal phase on cooling.

This is a meticulously reported study using X-ray diffraction, solid-state and solution NMR, DSC, dielectric spectroscopy, and DFT. The synthesis of the Ath+ cation is a significant synthetic achievement in its own right. A detailed experimental section provides the full detail needed to reproduce this work. In particular the crystallographic supporting information is all in order.

This work will certainly be of interest to the substantial molecular perovskite chemistry community, and should be published. On the other hand, I am not convinced that there is anything especially significant about the fact that the Ath+ material maintains a cubic space group. This is a zone-boundary transition like many others known in this family (see CrystEngComm, 2020, 22, 961-968; the Ath+ transition is of the "unconventional tilting" type with the X5- irrep, as shown in Fig 2 of that paper). The authors say that this is a "unique structural stability" (p. 9), but it is not clear in what sense this is either unique or stable: maintaining the same lattice system is not usually considered an indication of stability. Similarly, they suggest (p. 13) that this is somehow a more "pure order-disorder" transition than others, but it is not clear what this means. On the same page, they write that the A-site cation motion "can be largely decoupled from the perovskite matrix", but their own data show that this isn't correct: the large transverse ADPs on the azide ions in the HT phase, as well as the chemical expectation that these ions coordinate at an angle to Mn (p. 11), indicate that there is a substantial order-disorder transition in the framework too. (A better example of guest-matrix decoupling might be the R-3c - R3c transition in CH₃(NH₂)₂[Mn(HCO₂)₃], Chem. Mater. 2017, 29, 5, 2264-2275.)

For these reasons I recommend that the authors should revise their claims for the particular significance of this phase transition sequence. If there are alternative reasons why these materials are of broad interest, then this work might be appropriate for a general-interest journal; otherwise, perhaps it would be more suitable for a specialist chemistry journal.

Some more specific comments:

p. 3: Order-disorder phase transitions are not quite ubiquitous in the molecular perovskites: there are exceptions if there is strong host-guest hydrogen bonding, e.g., in the guanidinium formates.

Fig. 2: is the apparently split peak in the Q material at 350 K on heating evidence that the tetragonal phase is briefly formed on heating too?

p. 12: Is it really unusual to achieve a reversible, first-order transition between antiferroelectric and paraelectric phases above room T? This seems rather common among especially the azide perovskites.

Reviewer #3 (Remarks to the Author):

I think that this is a very interesting paper and deserves to be published in Nature Communications. The authors have designed a virtually spherical cage cation that can be accommodated in the A-site

cavity of a hybrid azide perovskite of general formula $AMn(N_3)_3$. It undergoes an unusual phase transition in which the structure is cubic in both the high temperature (paraelectric) and low temperature (antiferroelectric) phases. There is a change of space group, of course, as required for a 1st order transition. This transition, during which the azide framework remains virtually unchanged, has been carefully studied by single crystal X-ray diffraction, solid state NMR, calorimetry, and dielectric constant measurements.

The paper is very nicely written and the figures are very good. In addition, the quality of the work appears to be very thorough. I do, however, recommend that the references should be carefully reviewed for relevance. For example, the first FIFTEEN references concern halide perovskites such as $MAPbI_3$, which have been extensively studied in the recent literature due to their remarkable optoelectronic properties. On the other hand, these halide materials have relatively little in common with the authors' work on azide perovskites, which are more like the widely studied formates. I suppose that they were included because it enables the introduction to begin with references to a string of highly-cited Science, JACS and Nature X papers. I don't think that this is necessary because the work stands on its own merits. Furthermore, the authors have not referenced some of the more relevant work on azide perovskites with different amine cations. For example, it would be very interesting to compare the present phase transition with the unusual one described by Wei et al for the case where the amine is MA (see *Angewandte Chemie* (2018): doi.org/10.1002/ange.201803176). I strongly recommend that the authors revise the referencing to give more coverage of work on other azide perovskites, and much less on halides.

Aside from that, I noticed one minor omission, which was to state the counter anion in the solution NMR spectra shown in Figure 1.

A. K. Cheetham

General response to the reviewers:

We are grateful to all the reviewers for their insightful comments, suggestions, and compliments. We have refined our statements and inserted new analysis and discussions in the revised manuscript, highlighted in yellow, according to the reviewers' request. The quality of our work has been significantly enhanced, thanks to the reviewers' efforts.

REVIEWER 1

General comment:

Hybrid organic-inorganic perovskites are well-known compounds for their attractive properties that can be easily tuned by proper choice of organic and inorganic components. The authors report synthesis of a new azide perovskite comprising a new rigid and spheroidal organic cation. They show that the compound exhibits a structural phase transition from the $Pm-3m$ to $Pa-3$ phase. This transition is associated with change of the dielectric permittivity from about 6 to about 11. I have no doubts that the data are reliable and that analysis of almost all experimental data was performed in a correct way. I have, however, impression that the authors' conclusions are overclaimed. Therefore, I am not convinced that the paper is suitable for Nature Communications. It seems more suitable for a more specialized journal, like Chemistry of Materials or Journal of Materials Chemistry C.

Comment 1:

- Introduction: the authors wrote "This host-guest interaction has a strong influence on the macroscopic structure and properties of the perovskite material. For example, the transition of $FAPbI_3$ from the black perovskite phase to the yellow hexagonal phase is accompanied by FA^+ losing two degrees of its rotational freedom." The yellow phase is not a PEROVSKITE phase. Therefore, better example would be $MAPbI_3$, $MAPbBr_3$ or recently discovered methylhydrazinium lead bromide perovskites, which undergo phase transitions from the disordered to ordered phases.

Response:

Thanks for this suggestion. On p. 3, we have replaced this example by $MAPbI_3$, whose properties and stability are enhanced in the cubic phase:

"For example, $MAPbI_3$ is a distorted perovskite in the tetragonal crystal system at room temperature. When MA^+ is partially substituted by another cation, the resulting mixed-cation perovskite can take on the ideal cubic structure with significantly enhanced semiconducting properties and stability."

Comment 2:

- DSC data: the authors claim that "The cubic phase yields only a few diffraction points, insufficient for resolving the disordered A-site cation." Please add information on phase transition entropy and the calculated value of N using the formula $\Delta S = R \ln N$. This will

provide information on number of configurational states of the disordered cation in the HT phase.

Response:

This is an excellent point. We have inserted a new paragraph analyzing and discussing the entropy change during phase transition of $\text{AthMn}(\text{N}_3)_3$ on p. 9:

“From the DSC results, we estimated the entropy change of $\text{AthMn}(\text{N}_3)_3$ to be $15 \text{ J}\cdot\text{mol}^{-1}\cdot\text{K}^{-1}$ during its phase transition at 339–341 K. According to Boltzmann’s entropy formula

$$\Delta S = R \ln \frac{N_2}{N_1}$$

where N is the number of accessible microstates and R is the gas constant, a ratio of $N_2/N_1 = 6$ is obtained for the phase transition. This value is consistent with a simple mechanistic picture that Ath^+ gains a disorder/order ratio of 3 and the $\text{Mn}(\text{N}_3)_3^-$ matrix gains 2. By contrast, previously reported azide perovskites exhibited significantly higher ΔS values, such as $\sim 20 \text{ J}\cdot\text{mol}^{-1}\cdot\text{K}^{-1}$ for $\text{TMA Cd}(\text{N}_3)_3^{26}$ and $\text{TMAMn}(\text{N}_3)_3^{15}$, or $23\text{--}26 \text{ J}\cdot\text{mol}^{-1}\cdot\text{K}^{-1}$ for $\text{TMAM}^{I}_{0.5}\text{M}^{III}_{0.5}(\text{N}_3)_3^{28}$. Compared with these systems, the relatively small ΔS of $\text{AthMn}(\text{N}_3)_3$ suggests disorder in its HT phase to be well-defined, which highlights a close match of the $\text{Mn}(\text{N}_3)_3^-$ framework with the spheroidal profile of Ath^+ , more so than with other cations TMA^+ , Abh^+ , or Q^+ .”

These results further corroborates that $\text{AthMn}(\text{N}_3)_3$ studied herein is not a simple extension to existing systems, but a very special case.

Comment 3:

- *The authors wrote "the orientations of Ath^+ and N_3^- become disordered" and they conclude that " $\text{AthMn}(\text{N}_3)_3$ offers a prime model for studying the pure order–disorder transition and more disorder-related phenomena." They also wrote in the Abstract "Our work introduces a new family of perovskite structures and provides direct insights to the order–disorder transition of hybrid materials." and "This pure order–disorder transition..."*

Firstly, the phase transition leads also to tilting of the MnN_6 octahedra and I think that also to shift of the Ath^+ cation. These are displacive contributions and therefore the phase transition is not 100% of order-disorder type. I recommend to add some information on the shift of the organic cation.

Response:

Yes, we calculated the geometric center of Ath^+ in the LT phase to be $x = y = z = 0.27$, slightly shifted from the ideal situation $x = y = z = 0.25$. We have added the following information on p. 12:

“As a result of simultaneously forming hydrogen bonds ($\sim 2.4 \text{ \AA}$ in length) with

three N⁻ ligands, the Ath⁺ cage cation shifts slightly toward a Mn²⁺_{4b} ...”

Comment 4:

Secondly, the same symmetry change, from Pm-3m to Pa-3 associated with ordering of organic cation and N₃⁻ ligands were reported for double perovskite family (NMe₄)₂[M^{III}M^I(N₃)₆] (M^{III}=Cr, Fe; M^I=Na, K) (see Z.-Y. Du et al, Cryst. Growth Design. 2014, 14, 3906-3909). Therefore, novelty of this paper is limited and the authors cannot claim that their AthMn(N₃)₃ compound is unique and that it offers "a prime model for studying the pure order-disorder transition..."

- The authors should compare their results with those reported for the mentioned above tetramethylammonium double perovskite azides (Z.-Y. Du et al, Cryst. Growth Design. 2014, 14, 3906-3909).

Response:

Thanks for suggesting this excellent work to us. Indeed, TMA₂M^IM^{III}(N₃)₆ in this reference also undergoes an order–disorder phase transition, and we have cited this reference twice in the revised manuscript:

p. 4:

“The LT phases of several bimetallic perovskites TMA₂M^IM^{III}(N₃)₆ (M^I = Na, K; M^{III} = Cr, Fe) have been found²⁸ to adopt a cubic structure. Nonetheless, the nonpolarity of TMA⁺ is ineffective in inducing dielectric or ferroelectric switching during phase transitions.”

p. 9:

“From the DSC results, we estimated the entropy change of AthMn(N₃)₃ to be 15 J·mol⁻¹·K⁻¹ during its phase transition at 339–341 K. ... By contrast, previously reported azide perovskites exhibited significantly higher ΔS values, such as ~20 J·mol⁻¹·K⁻¹ for TMACd(N₃)₃²⁶ and TMAMn(N₃)₃¹⁵, or 23–26 J·mol⁻¹·K⁻¹ for TMAM^I_{0.5}M^{III}_{0.5}(N₃)₃.²⁸”

Nonetheless, there are significant differences between TMA₂M^IM^{III}(N₃)₆ in this reference and AthMn(N₃)₃ studied herein:

- The HT phase of bimetallic TMA₂M^IM^{III}(N₃)₆ adopts the Fm–3m space group, whereas AthMn(N₃)₃ adopts the Pm–3m space group;
- Ath⁺ in AthMn(N₃)₃ bears a permanent dipole that contributes to the dielectric behavior of AthMn(N₃)₃;
- Most importantly, the disorder of the perovskite framework of AthMn(N₃)₃ in the HT phase is limited, but TMA₂M^IM^{III}(N₃)₆ exhibits a large entropy change and substantial ionic character in the HT phase.

Comment 5:

- *The disorder of the azide anions is seen through very large thermal ellipsoid of the terminal N atoms. Can the structure be better refined assuming disordered model for the azide anions, like in the case of double perovskite azides?*

Response:

This is a very good question. Our current structural model places the central N of azide at the 3c position with a small and nearly isotropic thermal ellipsoid. The azide linker forms a *trans* bridge between two adjacent Mn²⁺ at disordered orientations (there may exist a local order between nearby azides). We found the large, oblate thermal ellipsoid at the 6f position to properly represent such kind of disorder for the azido N. Other disorder models give less accurate refining results, even though using more modeling parameters.

We have inserted the following explanation on p. 13:

“Still, it is presumable that the linear N₃⁻ maintains a *trans* configuration between adjacent Mn²⁺ at disordered orientations, represented by a nearly isotropic thermal ellipsoid in the middle and large, oblate ones for the terminal N⁻.”

Comment 6:

- *The authors wrote in the Abstract "The structure and properties of organic–inorganic hybrid perovskites are impacted by the order–disorder transition, whose driving forces from the organic cation and the inorganic framework cannot easily be disentangled." Other statements in the Abstract and the text suggest that in case of AthMn(N₃)₃ the driving force from the inorganic framework can be neglected. This sentence is a little bit misleading since although the framework structure remains cubic, this compounds still presents two driving forces that may be difficult to disentangled: one is ordering of the Ath⁺ cation and the second force is ordering of the N₃⁻ linkers. I have no doubt that the main driving force of the phase transition is ordering of the organic cation but for sure ordering of the azide anion also contributes to the phase transition mechanism.*

Response:

We totally agree with you that both Ath⁺ and Mn(N₃)₃⁻ contribute to the driving force of the phase transition. As mentioned in our response #2, we have made new quantitative analysis of the entropy change of phase transition on p. 9, indicating that Ath⁺ is main source of disorder:

“From the DSC results, we estimated the entropy change of AthMn(N₃)₃ to be 15 J·mol⁻¹·K⁻¹ during its phase transition at 339–341 K. According to Boltzmann’s entropy formula

$$\Delta S = R \ln \frac{N_2}{N_1}$$

where N is the number of accessible microstates and R is the gas constant, a ratio of $N_2/N_1 = 6$ is obtained for the phase transition. This value is consistent

with a simple mechanistic picture that Ath^+ gains a disorder/order ratio of 3 and the $\text{Mn}(\text{N}_3)_3^-$ matrix gains 2.”

We have also deleted the word "pure" and rewritten part of the conclusion to avoid possible confusions:

“Overall, thanks to the distinctive cage-in-framework structure of $\text{AthMn}(\text{N}_3)_3$, movement of the spheroidal Ath^+ directly contributes to the antiferroelectric–paraelectric transition and brings about little excess entropy to the perovskite matrix.”

REVIEWER 2

General comments:

- *The manuscript by Shi and co-workers presents three new members of the family of molecular perovskites, in which spheroidal polycyclic ammonium cations ("Abh⁺", "Ath⁺" and "Q⁺") sit at the A site within manganese azide cages. Like many other members of this family, the materials undergo an order-disorder, dielectric-antiferroelectric phase transition. The authors make the particular claim to significance that the Ath⁺ material has a cubic space group both above and below the phase transition -- unlike the Abh⁺ material, which has a low-temperature orthorhombic phase, and the Q⁺ material, which has an intriguing metastable tetragonal phase on cooling.*

This is a meticulously reported study using X-ray diffraction, solid-state and solution NMR, DSC, dielectric spectroscopy, and DFT. The synthesis of the Ath⁺ cation is a significant synthetic achievement in its own right. A detailed experimental section provides the full detail needed to reproduce this work. In particular the crystallographic supporting information is all in order.

Response:

Thank you very much for appreciating our work.

Comment 1:

-*This work will certainly be of interest to the substantial molecular perovskite chemistry community, and should be published. On the other hand, I am not convinced that there is anything especially significant about the fact that the Ath⁺ material maintains a cubic space group. This is a zone-boundary transition like many others known in this family (see CrystEngComm, 2020, 22, 961-968; the Ath⁺ transition is of the "unconventional tilting" type with the X₅⁻ irrep, as shown in Fig 2 of that paper).*

Response:

Thanks for this kind reminder. Using the ISODISTORT program, we identified the active tilting mode to be a roughly equal mixture of the X₅⁻ and M₅⁺ irreps. We have revised the corresponding sentence in p. 11 as follows:

"The ISODISTORT program³³ reveals that such unconventional tilting can be represented by a mixture of the X₅⁻ and M₅⁺ modes having similar amplitudes.³⁴"

Comment 2:

-*The authors say that this is a "unique structural stability" (p. 9), but it is not clear in what sense this is either unique or stable: maintaining the same lattice system is not usually considered an indication of stability. Similarly, they suggest (p. 13) that this is somehow a more "pure order-disorder" transition than others, but it is not clear what this means. On the same page, they write that the A-site cation motion "can be largely decoupled from the perovskite matrix", but their own data show that this isn't correct: the large transverse ADPs on the azide ions in the HT phase, as well as the chemical*

expectation that these ions coordinate at an angle to Mn (p. 11), indicate that there is a substantial order-disorder transition in the framework too. (A better example of guest-matrix decoupling might be the R-3c - R3c transition in $\text{CH}_3(\text{NH}_2)_2[\text{Mn}(\text{HCO}_2)_3]$, Chem. Mater. 2017, 29, 5, 2264–2275.)

For these reasons I recommend that the authors should revise their claims for the particular significance of this phase transition sequence. If there are alternative reasons why these materials are of broad interest, then this work might be appropriate for a general-interest journal; otherwise, perhaps it would be more suitable for a specialist chemistry journal.

Response:

Thanks very much for the general suggestions. We have deleted the word "pure" and rewritten part of the conclusion to avoid possible confusions:

“Overall, thanks to the distinctive cage-in-framework structure of $\text{AthMn}(\text{N}_3)_3$, movement of the spheroidal Ath^+ directly contributes to the antiferroelectric–paraelectric transition and brings about little excess entropy to the perovskite matrix.”

Moreover, we have carried out a quantitative study of the host–guest interaction. From the DSC results, we calculated the entropy change during phase transition to be $\Delta S = 15 \text{ J} \cdot \text{mol}^{-1} \cdot \text{K}^{-1}$, corresponding to an increase of 6 degrees of freedom. Compared with previously reported systems containing TMA^+ , our result is the smallest and closest to the theoretical value ($N = 3$). During the order–disorder transition, the marginal entropy gained by the perovskite framework reflects its stability or good host–guest match, more so than with other A-site cations. Therefore, we believe $\text{AthMn}(\text{N}_3)_3$ studied herein is not a simple extension to existing systems, but a very special case.

We have inserted a new paragraph analyzing and discussing the entropy change during phase transition of $\text{AthMn}(\text{N}_3)_3$ on p. 9:

“From the DSC results, we estimated the entropy change of $\text{AthMn}(\text{N}_3)_3$ to be $15 \text{ J} \cdot \text{mol}^{-1} \cdot \text{K}^{-1}$ during its phase transition at 339–341 K. ... Compared with these systems, the relatively small ΔS of $\text{AthMn}(\text{N}_3)_3$ suggests disorder in its HT phase to be well-defined, which highlights a close match of the $\text{Mn}(\text{N}_3)_3^-$ framework with the spheroidal profile of Ath^+ , more so than with other cations TMA^+ , Abh^+ , or Q^+ .”

Comment 3:

- p. 3: Order-disorder phase transitions are not quite ubiquitous in the molecular perovskites: there are exceptions if there is strong host-guest hydrogen bonding, e.g., in the guanidinium formates.

Response:

We agree with you on this point, and have replaced the word "ubiquitous" by "typical" in the revised manuscript.

Comment 4:

- Fig. 2: *is the apparently split peak in the Q material at 350 K on heating evidence that the tetragonal phase is briefly formed on heating too?*

Response:

This is a very good question. The endothermic peak for $\text{QMn}(\text{N}_3)_3$ at 350 K may well be two phase transition steps taking place continuously, except that this peak appears to be split into two disproportionate parts. We have revised the corresponding sentence on p. 9 as follows:

“The transition temperature of $\text{QMn}(\text{N}_3)_3$ is even higher at 351 K, possibly in two continuous steps, and the reverse transition occurs in separate steps at 347 and 320 K.”

Comment 5:

- p. 12: *Is it really unusual to achieve a reversible, first-order transition between antiferroelectric and paraelectric phases above room T? This seems rather common among especially the azide perovskites.*

Response:

Thanks for this query. Indeed, several azide perovskites, especially $\text{TMAMn}(\text{N}_3)_3$ and its homologs, are known to undergo order–disorder transitions around or above room temperature. However, the phase transition is not accompanied by a significant switch in the dielectric behavior. Therefore, we believe it is unusual for Ath^+ to drive the antiferroelectric–paraelectric transition of $\text{AthMn}(\text{N}_3)_3$.

We have revised the discussion on p. 13 as follows:

“This is distinct from $\text{TMAMn}(\text{N}_3)_3$ and its homologs,^{15,22} where the open-chain ammonium $(\text{CH}_3)_n\text{NH}_{4-n}^+$ at the A-site barely contributes to the dielectric behavior of the whole perovskite material. In short, the order–disorder transition of Ath^+ serves as the main driving force for $\text{AthMn}(\text{N}_3)_3$ to convert between the antiferroelectric and paraelectric phases.”

Reviewer 3

General Comment:

- I think that this is a very interesting paper and deserves to be published in Nature Communications. The authors have designed a virtually spherical cage cation that can be accommodated in the A-site cavity of a hybrid azide perovskite of general formula $AMn(N_3)_3$. It undergoes an unusual phase transition in which the structure is cubic in both the high temperature (paraelectric) and low temperature (antiferroelectric) phases. There is a change of space group, of course, as required for a 1st order transition. This transition, during which the azide framework remains virtually unchanged, has been carefully studied by single crystal X-ray diffraction, solid state NMR, calorimetry, and dielectric constant measurements.

- The paper is very nicely written and the figures are very good. In addition, the quality of the work appears to be very thorough.

Response:

Thanks a lot for these positive comments about our work.

Comment 1:

- I do, however, recommend that the references should be carefully reviewed for relevance. For example, the first FIFTEEN references concern halide perovskites such as $MAPbI_3$, which have been extensively studied in the recent literature due to their remarkable optoelectronic properties. On the other hand, these halide materials have relatively little in common with the authors' work on azide perovskites, which are more like the widely studied formates. I suppose that they were included because it enables the introduction to begin with references to a string of highly-cited Science, JACS and Nature X papers. I don't think that this is necessary because the work stands on its own merits. Furthermore, the authors have not referenced some of the more relevant work on azide perovskites with different amine cations. For example, it would be very interesting to compare the present phase transition with the unusual one described by Wei et al for the case where the amine is MA (see *Angewandte Chemie* (2018): doi.org/10.1002/ange.201803176). I strongly recommend that the authors revise the referencing to give more coverage of work on other azide perovskites, and much less on halides.

Response:

These are great suggestions. We have added several new literature on azide perovskites in our references:

26. Du, Z.-Y. et al. Above-room-temperature ferroelastic phase transition in a perovskite-like compound $[N(CH_3)_4][Cd(N_3)_3]$. *Chem. Commun.* **50**, 1989-1991 (2014).

28. Du, Z.-Y. et al. Structural transition in the perovskite-like bimetallic azido coordination polymers: $(NMe_4)_2[B'B''(N_3)_6]$ ($B' = Cr^{3+}, Fe^{3+}$; $B'' = Na^+, K^+$). *Cryst.*

Growth Des. **14**, 3903-3909 (2014).

34. Boström, H. L. B. Tilts and shifts in molecular perovskites.

CrystEngComm **22**, 961-968 (2020).

Comment 2:

-Aside from that, I noticed one minor omission, which was to state the counter anion in the solution NMR spectra shown in Figure 1.

Response:

Thanks for reminding us about this issue. We have included the counter anions in the caption of Fig. 1.

REVIEWERS' COMMENTS

Reviewer #1 (Remarks to the Author):

The authors revised the manuscript and I have no further comments. I recommend acceptance of this manuscript in its present form.

Mirosław Mączka

Reviewer #2 (Remarks to the Author):

I thank the authors for their detailed response to the referees' comments. I think they have addressed these well and I am pleased to recommend the manuscript for publication. The minor remarks below should be addressed before publication but don't require another round of review.

The one comment the authors have not addressed is referee 3's suggestion -- with which I concur -- that they remove most of the introductory references to MAPI and friends, which are more or less irrelevant to the current manuscript, and instead compare the system to related compounds such as the methylammonium manganese azide he refers to.

There is a typo in the x-axis label of figure 3 ("Chemical").

REVIEWER 1

Comment:

-The authors revised the manuscript and I have no further comments. I recommend acceptance of this manuscript in its present form.

Response:

Thank you very much for appreciating our work.

REVIEWER 2

General comment:

-I thank the authors for their detailed response to the referees' comments. I think they have addressed these well and I am pleased to recommend the manuscript for publication. The minor remarks below should be addressed before publication but don't require another round of review.

Comment 1:

-The one comment the authors have not addressed is referee 3's suggestion -- with which I concur -- that they remove most of the introductory references to MAPbI₃ and friends, which are more or less irrelevant to the current manuscript, and instead compare the system to related compounds such as the methylammonium manganese azide he refers to.

Response:

Thanks for your suggestion and we have removed several references about halide perovskites as below:

1. Zhou, H. *et al.* Interface engineering of highly efficient perovskite solar cells. *Science* **345**, 542-546 (2014).
3. Liu, M., Johnston, M. B. & Snaith, H. J. Efficient planar heterojunction perovskite solar cells by vapour deposition. *Nature* **501**, 395-398 (2013).
6. Liu, F. *et al.* Highly luminescent phase-stable CsPbI₃ perovskite quantum dots achieving near 100% absolute photoluminescence quantum yield. *ACS Nano* **11**, 10373-10383 (2017).
13. Liu, J. *et al.* High-quality mixed-organic-cation perovskites from a phase-pure non-stoichiometric intermediate (FAI)_{1-x}PbI₂ for solar cells. *Adv. Mater.* **27**, 4918-4923 (2015).
14. Shi, Z. *et al.* Symmetrization of the crystal lattice of MAPbI₃ boosts the performance and stability of metal-perovskite photodiodes. *Adv. Mater.* **29**, 1701656 (2017).

On p. 3, we have also replaced the MAPbI₃ example by DMAZn(HCOO)₃, which undergoes an order-disorder phase transition at ~156 K:

“For example, the phase transition of dimethylammonium zinc trifluoroacetate [DMAZn(HCOO)₃] from paraelectric to antiferroelectric at ~156 K is accompanied by DMA⁺ losing its rotational freedom.^{10,11”}

Comment 2:

-There is a typo in the x-axis label of figure 3 ("Chemical").

Response:

Thanks for this kind reminder. We have corrected this typo in the final version.